# Hierarchical Time Series Forecasting with Robust Reconciliation

**Shuhei Aikawa**  *aikawa.s.a647@m.isct.ac.jp*
*Department of Industrial Engineering and Economics*
*Institute of Science Tokyo*

**Aru Suzuki**  *suzuki.a.3382@m.isct.ac.jp*
*Department of Industrial Engineering and Economics*
*Institute of Science Tokyo*

**Kei Yoshitake**  *yoshitake.k.6312@m.isct.ac.jp*
*Department of Industrial Engineering and Economics*
*Institute of Science Tokyo*

**Kanata Teshigawara**  *teshigawara.k.46c4@m.isct.ac.jp*
*Department of Industrial Engineering and Economics*
*Institute of Science Tokyo*

**Akira Iwabuchi**  *iwabuchi.a.3c54@m.isct.ac.jp*
*Department of Industrial Engineering and Economics*
*Institute of Science Tokyo*

**Ken Kobayashi**  *kobayashi.k@iee.eng.isct.ac.jp*
*Department of Industrial Engineering and Economics*
*Institute of Science Tokyo*

**Kazuhide Nakata**  *nakata.k.ac@m.titech.ac.jp*
*Department of Industrial Engineering and Economics*
*Institute of Science Tokyo*

**Reviewed on OpenReview:** *https://openreview.net/forum?id=XHPLjF52gY*

## Abstract

This paper focuses on forecasting hierarchical time-series data, where each higher-level observation equals the sum of its corresponding lower-level time series. In such contexts, the forecast values should be coherent, meaning that the forecast value of each parent series exactly matches the sum of the forecast values of its child series. Existing hierarchical forecasting methods typically generate base forecasts independently for each series and then apply a reconciliation procedure to adjust them so that the resulting forecast values are coherent across the hierarchy. These methods generally yield an optimal reconciliation, using a covariance matrix of the forecast errors. In practice, however, the true covariance matrix is unknown and has to be estimated from finite samples in advance. This gap between the true and estimated covariance matrix may degrade forecast performance. To address this issue, we propose a robust optimization framework for hierarchical reconciliation that accounts for uncertainty in the estimated covariance matrix. We first introduce an uncertainty set for the estimated covariance matrix and formulate a reconciliation problem that minimizes the worst-case average of weighted squared residuals over this uncertainty set. We show that our problem can be cast as a semidefinite optimization problem. Numerical experiments demonstrate that the proposed robust reconciliation method achieved better forecast per-

formance than existing hierarchical forecasting methods, which indicates the effectiveness of integrating uncertainty into the reconciliation process.

# 1 Introduction

Time-series forecasting is indispensable across diverse fields, including sales planning and inventory management (Aviv, 2003; Ramos et al., 2015), energy supply planning (Suganthi & Samuel, 2012; Hernandez et al., 2014), and economic analysis and stock investment decision-making (Krollner et al., 2010). For instance, in the retail sector, accurate sales predictions based on historical data are crucial for optimizing inventory levels and preventing both overstocking and shortages. Similarly, for electric power companies, forecasting electricity consumption enables efficient facility operations and effective supply-demand balance management. Moreover, at both the individual and national levels, leveraging forecasts of economic indicators and stock prices can significantly contribute to wealth creation. Conversely, low forecast accuracy can lead to substantial losses and missed opportunities for individuals, businesses, and society as a whole. Consequently, extensive research has focused on developing various time-series forecasting methods, and there remains a strong demand for more precise techniques (Mahalakshmi et al., 2016; Liu et al., 2021; Wen et al., 2022).

Many real-world datasets inherently possess hierarchical structures. Examples include sales data organized by region or demographic statistics categorized by gender or age group, which are commonly recorded across multiple levels of aggregation. In practice, the appropriate hierarchical level for forecasting depends on the specific application, and this choice can significantly influence prediction outcomes. Generally, as one descends to lower hierarchical levels, the data becomes more granular but also more susceptible to individual variations and noise, leading to increased uncertainty. This often means that aggregated data at higher levels tends to be more stable and achieve greater forecast accuracy (Grunfeld & Griliches, 1960). However, higher-level aggregated data can obscure fine-grained patterns and individual variation factors. Therefore, it has also been suggested that utilizing detailed data from lower levels, if appropriately modeled, can potentially yield superior forecast accuracy (Orcutt et al., 1968; Edwards & Orcutt, 1969).

Given this context, time-series forecasting methods that explicitly account for hierarchical structures have garnered increasing attention (Athanasopoulos et al., 2009; Hyndman et al., 2011; Wickramasuriya et al., 2019; Shiratori et al., 2020; Panagiotelis et al., 2021; Hyndman & Athanasopoulos, 2021). These approaches aim to adjust forecasts across both lower and higher levels to ensure coherence when aggregating forecast values within the hierarchy. Such methods are expected to enhance forecast accuracy compared to conventional time-series forecasting based on a single level.

Despite these advancements, existing hierarchical time-series forecasting methods face certain challenges. Traditional approaches typically aim to minimize the expected forecast error at a given target time point, which necessitates a covariance matrix of the forecast errors. This matrix is commonly estimated from the residuals between observed and forecast values. However, if underlying data trends shift or the forecasting model is inaccurate, discrepancies can arise between the estimated and true covariance matrices. Thus, the estimated covariance matrix itself carries inherent uncertainty, which must be addressed. Prior work by Møller et al. (2024) focused on this issue, decomposing covariance matrix estimation into parameter estimation errors and stochastic irreducible errors to quantify uncertainty and improve forecast accuracy. Nevertheless, even with their method, the true covariance matrix cannot be perfectly determined, leaving room for further improvements in forecast accuracy.

To address the uncertainty inherent in estimators, robust optimization has emerged as a powerful technique. This methodology is designed to yield solutions that remain effective even when the underlying data fluctuates within a defined uncertain range. Specifically, it involves establishing a range for uncertain parameters or data and then seeking an optimal solution that performs best under the worst-case scenario within that range. Since its inception by Ben-Tal & Nemirovski (1998), robust optimization has been extensively researched in both theoretical and applied domains (Bertsimas et al., 2011). Notably, models that incorporate covariance matrix uncertainty have been developed and applied to various problems, such as portfolio optimization (Lobo & Boyd, 2000; Halldórsson & Tütüncü, 2003; Tütüncü & Koenig, 2004).

In this paper, we propose a novel method that frames hierarchical time-series forecasting as a robust optimization problem. Our approach aims to minimize the average of forecast residuals over an observation period under the worst-case scenario within an uncertainty set for the covariance matrix of forecast errors. We demonstrate through duality that this robust optimization problem can be formulated as a semidefinite optimization problem, which is theoretically solvable efficiently. Furthermore, we present numerical experiments on five real-world datasets, showing that our proposed method achieves more accurate forecasts than existing hierarchical time-series forecasting techniques.

## 2 Hierarchical time-series forecasting

### 2.1 Notation

This section defines the notation used throughout the paper, which is consistent with prior studies on hierarchical time-series forecasting (Athanasopoulos et al., 2009; Hyndman et al., 2011; Wickramasuriya et al., 2019; Panagiotelis et al., 2021; Hyndman & Athanasopoulos, 2021).

A hierarchical structure is defined by a series of nested levels. Level 0 is the fully aggregated series. Level 1 consists of the series obtained by disaggregating the Level 0 series, and Level 2 contains the series that further disaggregate each Level 1 series. This process continues until the bottom level, denoted as Level $K$, where its series can no longer be disaggregated.

Let $y_X^{(t)} \in \mathbb{R}$ denote the observation of a series $X$ at time $t$. The label $X$ is a series of labels representing the indices of each level. For example, a series $X$ that belongs to series $i$ at Level 1, series $j$ at Level 2, and series $k$ at Level 3 can be denoted by $ijk$. The series at Level 0 is simply written as $y^{(t)}$, without a series name $X$. A key property of hierarchical data is that, at any given time point, the value of a series at a specific level equals the sum of the values of the series nested directly below it:

$$y^{(t)} = \sum_i y_i^{(t)}, \; y_i^{(t)} = \sum_j y_{ij}^{(t)}, \; y_{ij}^{(t)} = \sum_k y_{ijk}^{(t)}.$$

To simplify the notation of the hierarchical structure, a matrix and vector expression is often used. Let $n$ be the total number of series and $m$ be the number of bottom-level series, which satisfy $n > m$. We denote the vector of all series observations at time $t$ as $\boldsymbol{y}^{(t)} \in \mathbb{R}^n$ and the vector of bottom-level observations as $\boldsymbol{b}^{(t)} \in \mathbb{R}^m$. With the summing matrix $\boldsymbol{S} \in \mathbb{R}^{n \times m}$ that dictates the way in which the bottom-level series aggregate, the hierarchical structure can be written as:

$$\boldsymbol{y}^{(t)} = \boldsymbol{S}\boldsymbol{b}^{(t)}. \tag{1}$$

When Equation (1) holds for the values of all series and the bottom-level series at each time $t$, it is said that the hierarchy is satisfied.

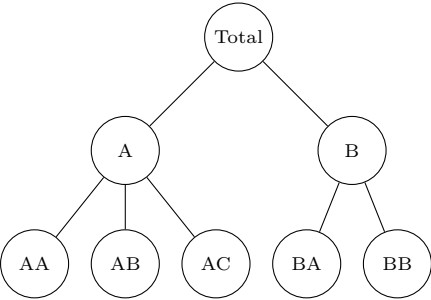

Figure 1: An example of a hierarchical structure

Figure 1 provides a simple example of a hierarchical structure. In this case, $K = 2$, $m = 5$, $n = 8$, and the following aggregation relationship holds:

$$y_{\mathrm{A}}^{(t)} = y_{\mathrm{AA}}^{(t)} + y_{\mathrm{AB}}^{(t)} + y_{\mathrm{AC}}^{(t)}, \; y_{\mathrm{B}}^{(t)} = y_{\mathrm{BA}}^{(t)} + y_{\mathrm{BB}}^{(t)},$$

$$y^{(t)} = y_{\mathrm{A}}^{(t)} + y_{\mathrm{B}}^{(t)} = y_{\mathrm{AA}}^{(t)} + y_{\mathrm{AB}}^{(t)} + y_{\mathrm{AC}}^{(t)} + y_{\mathrm{BA}}^{(t)} + y_{\mathrm{BB}}^{(t)}.$$

The corresponding matrix and vector representation is defined by:

$$\boldsymbol{y}^{(t)} = \left[ y^{(t)} \; y_{\mathrm{A}}^{(t)} \; y_{\mathrm{B}}^{(t)} \; y_{\mathrm{AA}}^{(t)} \; y_{\mathrm{AB}}^{(t)} \; y_{\mathrm{AC}}^{(t)} \; y_{\mathrm{BA}}^{(t)} \; y_{\mathrm{BB}}^{(t)} \right]^{\top},$$

$$\boldsymbol{b}^{(t)} = \left[ y_{\mathrm{AA}}^{(t)} \; y_{\mathrm{AB}}^{(t)} \; y_{\mathrm{AC}}^{(t)} \; y_{\mathrm{BA}}^{(t)} \; y_{\mathrm{BB}}^{(t)} \right]^{\top},$$

$$\boldsymbol{S} = \begin{bmatrix} 1 & 1 & 1 & 1 & 1 \\ 1 & 1 & 1 & 0 & 0 \\ 0 & 0 & 0 & 1 & 1 \\ 1 & 0 & 0 & 0 & 0 \\ 0 & 1 & 0 & 0 & 0 \\ 0 & 0 & 1 & 0 & 0 \\ 0 & 0 & 0 & 1 & 0 \\ 0 & 0 & 0 & 0 & 1 \end{bmatrix}.$$

With these definitions, the hierarchical structure is fully captured by Equation (1).

## 2.2 Reconciliation methods

Hierarchical time-series forecasting is a process that adjusts, or "reconciles," forecast values to ensure they satisfy the hierarchical aggregation constraints. We assume that observations for each series are available for an observation period $t = 1, \ldots, T$, and the objective is to forecast values for the forecast period $\tau = T+1, \ldots, T+T'$.

Forecasting, ignoring any aggregation constraints, is called a base forecast. Let $\hat{\boldsymbol{y}}^{(\tau)} \in \mathbb{R}^n$ denote the vector of base forecasts for all series at time $\tau$. A forecast that satisfies the hierarchical aggregation constraints is called a coherent forecast. Let $\tilde{\boldsymbol{y}}^{(\tau)} \in \mathbb{R}^n$ denote the vector of coherent forecasts at time $\tau$. To transform base forecasts into coherent forecasts, hierarchical forecasting methods estimate a reconciliation matrix $\boldsymbol{P} \in \mathbb{R}^{m \times n}$ that maps the base forecasts to the bottom level. All hierarchical forecasting approaches can be expressed in the general form:

$$\tilde{\boldsymbol{y}}^{(\tau)} = \boldsymbol{S}\boldsymbol{P}\hat{\boldsymbol{y}}^{(\tau)}, \tag{2}$$

where $\boldsymbol{S}$ is the summing matrix defined in Equation (1).

The bottom-up and top-down approaches are two traditional reconciliation methods. We again use the example from Figure 1 to illustrate these concepts. In the bottom-up approach, coherent forecasts are derived by summing the base forecasts of the bottom-level series. This corresponds to the reconciliation matrix:

$$\boldsymbol{P} = \begin{bmatrix} 0 & 0 & 0 & 1 & 0 & 0 & 0 & 0 \\ 0 & 0 & 0 & 0 & 1 & 0 & 0 & 0 \\ 0 & 0 & 0 & 0 & 0 & 1 & 0 & 0 \\ 0 & 0 & 0 & 0 & 0 & 0 & 1 & 0 \\ 0 & 0 & 0 & 0 & 0 & 0 & 0 & 1 \end{bmatrix}.$$

Conversely, in the top-down approach, coherent forecasts are obtained by disaggregating the base forecast of the top-level series. If $p_X$ is the proportion that allocates the total values to each bottom-level series $X$, the reconciliation matrix becomes:

$$\boldsymbol{P} = \begin{bmatrix} p_{\mathrm{AA}} & 0 & 0 & 0 & 0 & 0 & 0 & 0 \\ p_{\mathrm{AB}} & 0 & 0 & 0 & 0 & 0 & 0 & 0 \\ p_{\mathrm{AC}} & 0 & 0 & 0 & 0 & 0 & 0 & 0 \\ p_{\mathrm{BA}} & 0 & 0 & 0 & 0 & 0 & 0 & 0 \\ p_{\mathrm{BB}} & 0 & 0 & 0 & 0 & 0 & 0 & 0 \end{bmatrix}.$$

A common way to determine these proportions is based on the average historical proportions of the data.

Because the bottom-up and top-down methods utilize base forecasts from only a single level of aggregation, they rely on limited information. To overcome this, subsequent research has proposed methods that use base forecasts from all series to estimate a reconciliation matrix, thus producing more comprehensive, coherent forecasts.

Hyndman et al. (2011) proposed the generalized least squares (GLS) reconciliation, a regression-based approach. In this method, the reconciliation matrix is chosen to minimize errors between the base and coherent forecasts. Specifically, consider the regression model for base forecasts

$$\hat{\boldsymbol{y}}^{(\tau)} = \boldsymbol{S}\boldsymbol{\beta}^{(\tau)} + \boldsymbol{\varepsilon}^{(\tau)}.$$

Let $\boldsymbol{\beta}^{(\tau)} = \mathbb{E}\left[\boldsymbol{b}^{(\tau)} \mid \boldsymbol{y}^{(1)}, \ldots, \boldsymbol{y}^{(T)}\right] \in \mathbb{R}^m$ be the expectation of base forecasts at the bottom level, and let the error term $\boldsymbol{\varepsilon}^{(\tau)}$ have zero mean with covariance matrix $\boldsymbol{\Sigma}^{(\tau)} = \mathrm{Var}\left(\boldsymbol{\varepsilon}^{(\tau)} \mid \boldsymbol{y}^{(1)}, \ldots, \boldsymbol{y}^{(T)}\right) \in \mathbb{R}^{n \times n}$. If $\boldsymbol{\Sigma}^{(\tau)}$ were known, the minimum variance unbiased estimator of $\boldsymbol{\beta}^{(\tau)}$ would be the GLS estimator

$$\hat{\boldsymbol{\beta}}^{(\tau)} = \left(\boldsymbol{S}^\top \boldsymbol{\Sigma}^{(\tau)\dagger} \boldsymbol{S}\right)^{-1} \boldsymbol{S}^\top \boldsymbol{\Sigma}^{(\tau)\dagger} \hat{\boldsymbol{y}}^{(\tau)},$$

where $\boldsymbol{\Sigma}^{(\tau)\dagger}$ is the generalized inverse of $\boldsymbol{\Sigma}^{(\tau)}$. Comparing this expression with Equation (2) gives

$$\boldsymbol{P} = \left(\boldsymbol{S}^\top \boldsymbol{\Sigma}^{(\tau)\dagger} \boldsymbol{S}\right)^{-1} \boldsymbol{S}^\top \boldsymbol{\Sigma}^{(\tau)\dagger}.$$

In practice, however, the covariance matrix $\boldsymbol{\Sigma}^{(\tau)}$ is unknown and cannot be estimated. It represents the covariance of the reconciliation errors at time $\tau$, but the errors are not observable until coherent forecasts have already been produced. Hyndman et al. (2011) showed that, under the assumption that the errors themselves satisfy the aggregation constraints, $\boldsymbol{\Sigma}^{(\tau)}$ can be replaced with the $n$-dimensional identity matrix $\boldsymbol{I}_n$. This replacement is equivalent to computing the ordinary least squares (OLS) estimator instead of the GLS estimator.

Wickramasuriya et al. (2019) proposed the minimum-trace (MinT) reconciliation, which determines the reconciliation matrix $\boldsymbol{P}$ by minimizing the total variance of errors between the observed values and the coherent forecasts. Let

$$\boldsymbol{V}^{(\tau)} = \mathrm{Var}\left(\boldsymbol{y}^{(\tau)} - \tilde{\boldsymbol{y}}^{(\tau)} \mid \boldsymbol{y}^{(1)}, \ldots, \boldsymbol{y}^{(T)}\right) \in \mathbb{R}^{n \times n}$$

denote the covariance matrix of those errors, and define the covariance matrix of the errors between observed values and base forecasts as

$$\boldsymbol{W}^{(\tau)} = \mathrm{Var}\left(\boldsymbol{y}^{(\tau)} - \hat{\boldsymbol{y}}^{(\tau)} \mid \boldsymbol{y}^{(1)}, \ldots, \boldsymbol{y}^{(T)}\right) \in \mathbb{R}^{n \times n}.$$

Under the linear transformation in Equation (2), the error covariance matrix of the coherent forecasts is expressed as $\boldsymbol{V}^{(\tau)} = \boldsymbol{S}\boldsymbol{P}\boldsymbol{W}^{(\tau)}\boldsymbol{P}^\top \boldsymbol{S}^\top$. Consequently, the optimal reconciliation matrix that minimizes the total error variance $\mathrm{tr}\left(\boldsymbol{V}^{(\tau)}\right)$ is given by

$$\boldsymbol{P} = \left(\boldsymbol{S}^\top \left(\boldsymbol{W}^{(\tau)}\right)^{-1} \boldsymbol{S}\right)^{-1} \boldsymbol{S}^\top \left(\boldsymbol{W}^{(\tau)}\right)^{-1}. \tag{3}$$

A notable property of this reconciliation is that the resulting coherent forecasts $\tilde{\boldsymbol{y}}^{(\tau)}$ are guaranteed to be at least as accurate as the base forecasts $\hat{\boldsymbol{y}}^{(\tau)}$ in terms of the weighted squared error using $\left(\boldsymbol{W}^{(\tau)}\right)^{-1}$:

$$\left(\boldsymbol{y}^{(\tau)} - \tilde{\boldsymbol{y}}^{(\tau)}\right)^\top \left(\boldsymbol{W}^{(\tau)}\right)^{-1} \left(\boldsymbol{y}^{(\tau)} - \tilde{\boldsymbol{y}}^{(\tau)}\right) \le \left(\boldsymbol{y}^{(\tau)} - \hat{\boldsymbol{y}}^{(\tau)}\right)^\top \left(\boldsymbol{W}^{(\tau)}\right)^{-1} \left(\boldsymbol{y}^{(\tau)} - \hat{\boldsymbol{y}}^{(\tau)}\right).$$

As noted by Panagiotelis et al. (2021), when the base forecast $\hat{\boldsymbol{y}}^{(\tau)}$ is unbiased, the trace minimization in MinT is equivalent to minimizing the expected squared Euclidean norm of the errors between the observed values and the coherent forecasts:

$$\min_{\boldsymbol{P}} \quad \mathbb{E}[\|\boldsymbol{y}^{(\tau)} - \boldsymbol{S}\boldsymbol{P}\hat{\boldsymbol{y}}^{(\tau)}\|^2], \tag{4}$$

where $\|\cdot\|$ is the Euclidean norm. Furthermore, they provided a key insight that extends this Euclidean norm to a more general weighted norm. Specifically, they showed that if we know the true covariance matrix $\boldsymbol{W}^{(\tau)}$, the solution in Equation (3) is invariant to the choice of the weight matrix $\boldsymbol{M}$ in the following generalized optimization problem:

$$\min_{\boldsymbol{P}} \quad \mathbb{E}\left[(\boldsymbol{y}^{(\tau)} - \boldsymbol{SP}\hat{\boldsymbol{y}}^{(\tau)})^\top \boldsymbol{M}(\boldsymbol{y}^{(\tau)} - \boldsymbol{SP}\hat{\boldsymbol{y}}^{(\tau)})\right],\tag{5}$$

where $\boldsymbol{M}$ is any positive definite matrix. This property implies that the specific weighting of errors does not alter the optimal reconciliation matrix $\boldsymbol{P}$.

## 3 Robust reconciliation

The invariance property of MinT is a key advantage. At the same time, applying this to real-world data often involves practical considerations. For example, base forecasts may contain some empirical biases. Furthermore, because the true error distribution is unknown, we typically replace the expectation in Equation (5) with the average of residuals over a specific observation period $t = 1, \ldots, T$. In such cases, the choice of the weight matrix $\boldsymbol{M}$ can influence the performance of reconciliation.

In MinT framework, it is natural to use $(\boldsymbol{W}^{(\tau)})^{-1}$ as the weight matrix $\boldsymbol{M}$ to account for the different scales of errors across series. Although the true covariance matrix $\boldsymbol{W}^{(\tau)}$ is unknown, it is common to estimate it using the observations and base forecasts from the observation period $t = 1, \ldots, T$. This estimate is usually assumed to be constant for all forecast periods, so we consider using an estimated covariance matrix $\boldsymbol{W}$ as an alternative to $\boldsymbol{W}^{(\tau)}$ for all $\tau = T + 1, \ldots, T + T'$.

However, this single estimate $\boldsymbol{W}$ may not always be accurate due to limited data or changes in the underlying error structure. To address this uncertainty, we propose a robust optimization framework. We treat $\boldsymbol{W}^{-1}$ as an uncertain parameter and introduce an uncertainty set. By formulating a problem that minimizes the worst-case error over this uncertainty set, we aim to ensure stable forecast performance even when the estimate of $\boldsymbol{W}$ is not perfect.

### 3.1 Formulation

As we focus on the empirical average over $T$ time steps from the observation period, the minimization problem of weighted squared residuals is formulated as follows:

$$\min_{\boldsymbol{P}} \quad \frac{1}{T}\sum_{t=1}^{T}\left(\boldsymbol{y}^{(t)} - \boldsymbol{SP}\hat{\boldsymbol{y}}^{(t)}\right)^\top \boldsymbol{W}^{-1}\left(\boldsymbol{y}^{(t)} - \boldsymbol{SP}\hat{\boldsymbol{y}}^{(t)}\right),\tag{6}$$

where $\hat{\boldsymbol{y}}^{(t)}$ is the base forecast at time $t$ in the observation period. To incorporate robustness against the uncertainty in estimating $\boldsymbol{W}$, we introduce an uncertainty set $\mathcal{M} \subseteq \mathbb{R}^{n \times n}$. Then, we consider the following robust optimization problem:

$$\min_{\boldsymbol{P}} \max_{\boldsymbol{M} \in \mathcal{M}} \quad \sum_{t=1}^{T}\left(\boldsymbol{y}^{(t)} - \boldsymbol{SP}\hat{\boldsymbol{y}}^{(t)}\right)^\top \boldsymbol{M}\left(\boldsymbol{y}^{(t)} - \boldsymbol{SP}\hat{\boldsymbol{y}}^{(t)}\right),$$

where we omit the coefficient $1/T$ from the objective function, since it does not affect the optimal solution.

In robust optimization problems with uncertainty in the covariance matrix, a box uncertainty set is often used (Lobo & Boyd, 2000; Halldórsson & Tütüncü, 2003). The box uncertainty set places upper and lower bounds on the inverse of covariance matrix, so that

$$\mathcal{M} = \left\{\boldsymbol{M} \mid \underline{\boldsymbol{M}} \leq \boldsymbol{M} \leq \overline{\boldsymbol{M}}, \ \boldsymbol{M} \succeq \boldsymbol{O}\right\},$$

where $\overline{\boldsymbol{M}}, \underline{\boldsymbol{M}} \in \mathbb{R}^{n \times n}$ are the upper and lower bounds of $\boldsymbol{M}$, respectively, and the inequalities $\underline{\boldsymbol{M}} \leq \boldsymbol{M} \leq \overline{\boldsymbol{M}}$ represent element-wise inequalities, meaning $\underline{M}_{ij} \leq M_{ij} \leq \overline{M}_{ij}$ for all $i, j = 1, \ldots, n$. $\boldsymbol{M} \succeq \boldsymbol{O}$ denotes that

$\boldsymbol{M}$ is positive semidefinite. Therefore, the proposed method determines the reconciliation matrix by solving the following robust optimization problem:

$$\min_{\boldsymbol{P}} \max_{\boldsymbol{M}} \quad \sum_{t=1}^{T} \left(\boldsymbol{y}^{(t)} - \boldsymbol{SP}\hat{\boldsymbol{y}}^{(t)}\right)^{\top} \boldsymbol{M} \left(\boldsymbol{y}^{(t)} - \boldsymbol{SP}\hat{\boldsymbol{y}}^{(t)}\right)$$
$$\text{s.t.} \quad \underline{\boldsymbol{M}} \le \boldsymbol{M} \le \overline{\boldsymbol{M}}, \tag{7}$$
$$\boldsymbol{M} \succeq \boldsymbol{O}.$$

### 3.2 Equivalent reformulation

We show that the min-max problem (7) is equivalently reformulated as a semidefinite optimization problem following the approach of Lobo & Boyd (2000) and Halldórsson & Tütüncü (2003). Let $\boldsymbol{I}$ and $\boldsymbol{O}$ be the identity matrix and zero matrix of appropriate dimensions, respectively. For two symmetric matrices $\boldsymbol{A}$ and $\boldsymbol{B}$ of the same size, $\boldsymbol{A} \bullet \boldsymbol{B}$ denotes the standard inner product of the two matrices, defined as $\boldsymbol{A} \bullet \boldsymbol{B} := \text{tr}\left(\boldsymbol{A}^{\top}\boldsymbol{B}\right)$, which is the sum of the element-wise product of $\boldsymbol{A}$ and $\boldsymbol{B}$.

**Proposition 1.** *Assume that there exists a positive definite matrix $\boldsymbol{M}'$ satisfying $\underline{\boldsymbol{M}} < \boldsymbol{M}' < \overline{\boldsymbol{M}}$. Then, the problem* (7) *can be equivalently reformulated as the following semidefinite optimization problem:*

$$\min_{\overline{\boldsymbol{X}}, \underline{\boldsymbol{X}}, \boldsymbol{E}, \boldsymbol{P}} \quad \overline{\boldsymbol{M}} \bullet \overline{\boldsymbol{X}} - \underline{\boldsymbol{M}} \bullet \underline{\boldsymbol{X}}$$
$$\text{s.t.} \quad \begin{bmatrix} \overline{\boldsymbol{X}} - \underline{\boldsymbol{X}} & \boldsymbol{E} \\ \boldsymbol{E}^{\top} & \boldsymbol{I} \end{bmatrix} \succeq \boldsymbol{O}, \tag{8}$$
$$\boldsymbol{E} = \left[\boldsymbol{y}^{(1)} - \boldsymbol{SP}\hat{\boldsymbol{y}}^{(1)}, \dots, \boldsymbol{y}^{(T)} - \boldsymbol{SP}\hat{\boldsymbol{y}}^{(T)}\right],$$
$$\overline{\boldsymbol{X}}, \underline{\boldsymbol{X}} \ge \boldsymbol{O},$$

*where $\overline{\boldsymbol{X}}$ and $\underline{\boldsymbol{X}}$ are $n$-dimensional symmetric matrix variables, and $\boldsymbol{E}$ is an $n \times T$-dimensional matrix variable whose columns are the error vectors between the observed values and coherent forecasts over the observation period.*

*Proof.* Fix $\boldsymbol{P}$ and consider the inner maximization of the problem (7):

$$\max_{\boldsymbol{M}} \quad \sum_{t=1}^{T} \left(\boldsymbol{y}^{(t)} - \boldsymbol{SP}\hat{\boldsymbol{y}}^{(t)}\right)^{\top} \boldsymbol{M} \left(\boldsymbol{y}^{(t)} - \boldsymbol{SP}\hat{\boldsymbol{y}}^{(t)}\right)$$
$$\text{s.t.} \quad \underline{\boldsymbol{M}} \le \boldsymbol{M} \le \overline{\boldsymbol{M}}, \tag{9}$$
$$\boldsymbol{M} \succeq \boldsymbol{O}.$$

Its dual problem is formulated as

$$\min_{\overline{\boldsymbol{X}}, \underline{\boldsymbol{X}}} \quad \overline{\boldsymbol{M}} \bullet \overline{\boldsymbol{X}} - \underline{\boldsymbol{M}} \bullet \underline{\boldsymbol{X}}$$
$$\text{s.t.} \quad \overline{\boldsymbol{X}} - \underline{\boldsymbol{X}} - \sum_{t=1}^{T} \left(\boldsymbol{y}^{(t)} - \boldsymbol{SP}\hat{\boldsymbol{y}}^{(t)}\right) \left(\boldsymbol{y}^{(t)} - \boldsymbol{SP}\hat{\boldsymbol{y}}^{(t)}\right)^{\top} \succeq \boldsymbol{O}, \tag{10}$$
$$\overline{\boldsymbol{X}}, \underline{\boldsymbol{X}} \ge \boldsymbol{O},$$

where $\overline{\boldsymbol{X}}, \underline{\boldsymbol{X}}$ are dual variables. Let us define the matrix $\boldsymbol{E} = \left[\boldsymbol{y}^{(1)} - \boldsymbol{SP}\hat{\boldsymbol{y}}^{(1)}, \dots, \boldsymbol{y}^{(T)} - \boldsymbol{SP}\hat{\boldsymbol{y}}^{(T)}\right]$. Then, by the Schur complement, the semidefinite constraint in the problem (10) is equivalently transformed as follows:

$$\overline{\boldsymbol{X}} - \underline{\boldsymbol{X}} - \sum_{t=1}^{T} \left(\boldsymbol{y}^{(t)} - \boldsymbol{SP}\hat{\boldsymbol{y}}^{(t)}\right) \left(\boldsymbol{y}^{(t)} - \boldsymbol{SP}\hat{\boldsymbol{y}}^{(t)}\right)^{\top} \succeq \boldsymbol{O} \iff \overline{\boldsymbol{X}} - \underline{\boldsymbol{X}} - \boldsymbol{E}\boldsymbol{E}^{\top} \succeq \boldsymbol{O}$$
$$\iff \begin{bmatrix} \overline{\boldsymbol{X}} - \underline{\boldsymbol{X}} & \boldsymbol{E} \\ \boldsymbol{E}^{\top} & \boldsymbol{I} \end{bmatrix} \succeq \boldsymbol{O}. \tag{11}$$

Therefore, we can replace the semidefinite constraint of the problem (10) with Equation (11), which yields the following equivalent dual problem:

$$
\begin{aligned}
\min_{\overline{X}, \underline{X}} \quad & \overline{M} \bullet \overline{X} - \underline{M} \bullet \underline{X} \\
\text{s.t.} \quad & \begin{bmatrix} \overline{X} - \underline{X} & E \\ E^\top & I \end{bmatrix} \succeq O, \\
& \overline{X}, \ \underline{X} \geq O.
\end{aligned}
\tag{12}
$$

Finally, we show the dual problem (12) has a strictly feasible solution, which implies that the strong duality between the problems (9) and (12) holds. Let $J \in \mathbb{R}^{n \times n}$ be a matrix whose elements are all one. For the fixed $P$, we consider the following solution:

$$
\overline{X}' = EE^\top + \gamma J + I, \quad \underline{X}' = \gamma J,
$$

where $\gamma > 0$ is a sufficiently large constant such that $EE^\top + \gamma J > O$. Since we take $\gamma > 0$ sufficiently large, $\overline{X}', \underline{X}' > O$. Focusing on the semidefinite constraint of the problem (12), we have

$$
\begin{bmatrix} \overline{X}' - \underline{X}' & E \\ E^\top & I \end{bmatrix} = \begin{bmatrix} EE^\top + I & E \\ E^\top & I \end{bmatrix} \succ O,
$$

thus $\left( \overline{X}', \underline{X}' \right)$ is strictly feasible. Since we assume that the primal problem (9) is strictly feasible, the strong duality theorem holds and the optimal values of the two problems (9) and (12) are equal (Vandenberghe & Boyd, 1996). As we took $P$ arbitrarily, the above argument holds for all $P$. Therefore, the min-max problem (7) is equivalently reformulated as the semidefinite optimization problem (8). □

Hence, the approach of the proposed method is to solve the semidefinite optimization problem (8) and use the optimal solution for $P$ as the reconciliation matrix for hierarchical time-series forecasts.

### 3.3 Uncertainty set

In order to deal with the robust optimization problem described in the previous sections, it is necessary to determine the uncertainty set, i.e., $\overline{M}$ and $\underline{M}$ in advance. We set the upper and lower bounds of the uncertainty set from the observation period data using bootstrap with reference to Bertsimas et al. (2018).

We summarize the method for determining the uncertainty set in Algorithm 1. In the first step, we calculate a parameter $\lambda$ for the shrinkage approach, similar to the existing hierarchical time-series forecasting described in Section 2.2. This shrinkage approach is applied to the inverse of covariance matrix estimated by unbiased variance from the observation period data. In the next step, we estimate the inverse of covariance matrix of each sample obtained by sampling the data of the observation period. For sampling, we select the same number of time points as the observation period $T$ with replacement. Then, using the shrinkage intensity parameter $\lambda$ obtained in the first step, the shrinkage approach is applied to the sampled inverse covariance matrix. This sampling is repeated $N_B \geq 1$ times to obtain $N_B$ inverse covariance matrices. In the last step, determine upper and lower bounds from each element of the sampled inverse covariance matrices. Let $0 < \alpha \leq 1$, then set the width of the uncertainty set to be $\alpha$ times the width of the maximum and minimum sample values of the inverse of covariance matrix.

## 4 Numerical experiments

To verify the effectiveness of the proposed method, we compared its forecast accuracy with that of existing hierarchical time-series forecasting methods across multiple real-world datasets. Code used in these experiments is available at `https://github.com/isct-nakatalab/hierarchical-tsf-robust-reconciliation`.

---

**Algorithm 1** Designing uncertainty set

---

**Input:** $n, T, \{\boldsymbol{y}^{(t)}\}_{t=1}^T, \{\hat{\boldsymbol{y}}^{(t)}\}_{t=1}^T, N_B, \alpha$
**Output:** $\overline{\boldsymbol{M}}, \underline{\boldsymbol{M}}$
   $\boldsymbol{M} \leftarrow$ estimate inverse of covariance matrix from $\{\boldsymbol{y}^{(t)}\}_{t=1}^T$ and $\{\hat{\boldsymbol{y}}^{(t)}\}_{t=1}^T$
   $\lambda \leftarrow$ calculate shrinkage intensity parameter of $\boldsymbol{M}$
   **for** $s = 1, \ldots, N_B$ **do**
      $\{\boldsymbol{y}^{(t')}\}_{t'=1}^T, \{\hat{\boldsymbol{y}}^{(t')}\}_{t'=1}^T \leftarrow$ sample $T$ data points from $\{\boldsymbol{y}^{(t)}\}_{t=1}^T$ and $\{\hat{\boldsymbol{y}}^{(t)}\}_{t=1}^T$ with replacement
      $\boldsymbol{M}_{(s)} \leftarrow$ estimate inverse of covariance matrix from $\{\boldsymbol{y}^{(t')}\}_{t'=1}^T$ and $\{\hat{\boldsymbol{y}}^{(t')}\}_{t'=1}^T$
      $\boldsymbol{M}_{(s)} \leftarrow$ update $\boldsymbol{M}_{(s)}$ by shrinkage estimator with $\lambda$
   **end for**
   **for** $i = 1, \ldots, n, \ j = 1, \ldots, n$ **do**
      $\overline{M}_{ij} \leftarrow$ calculate $100 \cdot (1 + \alpha)/2$-percentile of $\{M_{(s)ij}\}_{s=1}^{N_B}$
      $\underline{M}_{ij} \leftarrow$ calculate $100 \cdot (1 - \alpha)/2$-percentile of $\{M_{(s)ij}\}_{s=1}^{N_B}$
   **end for**
   **return** $\overline{\boldsymbol{M}}, \underline{\boldsymbol{M}}$

---

## 4.1 Datasets

The numerical experiments used four datasets from prior studies. An overview of each dataset is provided below, including the number of hierarchical levels and series, as well as the lengths of the observation and forecast periods. These four datasets allowed us to assess the proposed method's performance under various real-world conditions, including hierarchies with different scales of levels and series.

The first dataset is the Australian births data (Hyndman & Athanasopoulos, 2021). It records the number of births in Australia every month from January 1975 to December 2022. As summarized in Table 1, the hierarchy consisted of a single disaggregation level ($K = 1$) with nine bottom-level series ($m = 9$) and ten series in total ($n = 10$). Level 1, the bottom level, disaggregated the national series into nine states and territories: ACT, AUS, NSW, NT, QLD, SA, TAS, VIC, and WA. For this experiment, the first 516 months (January 1975 to December 2017) served as the observation period, while the following 60 months (January 2018 to November 2022) comprised the forecast period.

Table 1: Hierarchy for Australian births data

| Level | Number of series | Total series per level |
|-------|:---:|:---:|
| total | 1 | 1 |
| state | 9 | 9 |

The second dataset is the Australian tourism data (Athanasopoulos et al., 2009), which records the number of domestic travelers every quarter from Q1 1998 to Q4 2017. As Table 2 shows, the hierarchy was structured with $K = 2$, $m = 27$, and $n = 35$. Level 1 disaggregated the national total into seven states: NSW, NT, QLD, SA, TAS, VIC, and WA. Note that this state grouping differed from the one used for the births dataset, as we followed prior work. Level 2, the bottom level, further broke down each state into finer geographic zones. Here, the observation period consisted of the first 68 quarters (Q1 1998 to Q4 2014), and the forecast period covered the subsequent 12 quarters (Q1 2015 to Q4 2017).

Table 2: Hierarchy for Australian tourism data

| Level | Number of series | Total series per level |
|-------|:---:|:---:|
| total | 1 | 1 |
| state | 7 | 7 |
| zone | 6-2-4-4-3-5-3 | 27 |

The third dataset is the U.S. Walmart sales data (Mancuso et al., 2021). It tracks weekly sales from January 3, 2011, to May 29, 2016. Table 3 shows the hierarchy as $K = 2$, $m = 10$, and $n = 14$. Level 1 split the national total into three states: CA, TX, and WI. Level 2, the bottom level, further subdivided each state into its constituent stores: four in CA, three in TX, and three in WI. In this experiment, we used the first 261 weeks (January 3, 2011, to January 3, 2016) as the observation period and the next 21 weeks (January 4, 2016, to May 29, 2016) for the forecast period.

Table 3: Hierarchy for Walmart sales data

| Level | Number of series | Total series per level |
|-------|------------------|------------------------|
| total | 1 | 1 |
| state | 3 | 3 |
| store | 4-3-3 | 10 |

The fourth dataset is the Swiss electricity demand data (Nespoli et al., 2020), recording electricity supply every ten minutes from January 13, 2018, to January 19, 2019. As outlined in Table 4, the hierarchy was $K = 3$, $m = 24$, and $n = 31$. Level 1 disaggregated the grid into two synthetic meter aggregations: S1 and S2. Level 2 further subdivided each aggregation into two synthetic sub-aggregations: S11 and S12 for S1, and S21 and S22 for S2. Level 3, the bottom level, then separated each sub-aggregation into six individual meters. After converting the data to a daily unit, we used the first 353 days (January 13, 2018, to December 31, 2018) for the observation period and the subsequent 19 days (January 1, 2019, to January 19, 2019) for the forecast period.

Table 4: Hierarchy for Swiss electricity demand data

| Level | Number of series | Total series per level |
|-------|------------------|------------------------|
| grid | 1 | 1 |
| agg. | 2 | 2 |
| sub-agg. | 2-2 | 4 |
| meter | 6-6-6-6 | 24 |

## 4.2 Experimental setup

This subsection details the benchmark methods, our proposed robust methods, and evaluation metrics.

Hierarchical forecasting first requires a set of base forecasts (**Base**), which do not account for the hierarchical structure. We generated these base forecasts using Prophet (Taylor & Letham, 2018), an open-source time-series library from Meta, with its default settings. As a preliminary experiment, we also performed base forecasts using other time-series forecasting methods such as ARIMA and LightGBM, but Prophet showed the best results. Our comparative methods included the bottom-up (**BU**), top-down (**TD**), GLS reconciliation (**GLS**), and MinT reconciliation (**MinT**) approaches introduced in Section 2.

For our proposed method (**Robust**), the optimization problem (8) was solved using the MOSEK solver (MOSEK ApS, 2025). Moreover, we needed to set two parameters for the bootstrap to design the uncertainty set, the number of sampling $N_B$ and the width of the uncertainty set $\alpha$. In these experiments, $N_B$ was fixed at 5000 and $\alpha$ was determined by validation from among the candidates specified in advance. For validation, we divided the observation period data into train and validation data in a ratio of 9:1, and used the $\alpha$ with the smallest RMSE in the validation data. The candidates for $\alpha$ were 0.5, 0.6, 0.7, 0.8, 0.9, and 1.0.

As evaluation metrics, we used the mean absolute error (MAE) and the root mean squared error (RMSE) for each series. For a given series $X$, these are defined as:

$$\text{MAE} = \frac{1}{T'} \sum_{\tau=T+1}^{T+T'} \left| y_X^{(\tau)} - \tilde{y}_X^{(\tau)} \right|, \ \text{RMSE} = \sqrt{\frac{1}{T'} \sum_{\tau=T+1}^{T+T'} \left( y_X^{(\tau)} - \tilde{y}_X^{(\tau)} \right)^2}$$

Note that RMSE gives a harsh evaluation when the prediction deviates significantly from the MAE.

### 4.3 Results and discussion

The experimental results for all datasets are summarized in Table 5–8. To make the results easier to understand, we calculated the ratio of MAE and RMSE for each hierarchical time-series forecasting method when the MAE and RMSE of the base forecasts are set to 1, and then calculated the mean and standard deviation for series included in the same hierarchical level. We defined these metrics as relative MAE and relative RMSE, respectively. The original MAE and RMSE of all series are listed in the appendix. In the tables, the format is "mean ± standard deviation." Since the top level of hierarchical structure (Level 0) includes only one series, the standard deviation for the series is not shown. The underlined values indicate the best forecast accuracy in terms of the mean for the corresponding hierarchical level.

For the Australian births dataset, the validation to decide a parameter for bootstrap resulted in $\alpha = 1.0$. The proposed method achieved the highest accuracy in most cases. In the top level, our proposed method was the only approach that achieved higher accuracy than the base forecast. In the bottom level, considering the standard deviation, some series showed significant improvement in forecast accuracy with the proposed method.

Table 5: Forecast accuracy for Australian births data

(a) Relative MAE

|  | Base | BU | TD | GLS | MinT | Robust |
|---|---|---|---|---|---|---|
| total | 1.000 | 1.034 | 1.000 | 1.003 | 0.999 | 0.923 |
| state | 1.000 | $1.000 \pm 0.000$ | $1.513 \pm 1.302$ | $0.955 \pm 0.136$ | $0.978 \pm 0.043$ | $0.944 \pm 0.167$ |

(b) Relative RMSE

|  | Base | BU | TD | GLS | MinT | Robust |
|---|---|---|---|---|---|---|
| total | 1.000 | 1.014 | 1.000 | 1.001 | 1.000 | 0.968 |
| state | 1.000 | $1.000 \pm 0.000$ | $1.297 \pm 0.831$ | $0.970 \pm 0.090$ | $0.989 \pm 0.026$ | $0.973 \pm 0.103$ |

In the Australian tourism dataset, the bootstrap parameter was determined via validation to be $\alpha = 0.5$. Our proposed method yielded lower forecast accuracy than the existing GLS and MinT reconciliation methods, except for the top level. This underperformance seems to be due to a slight discrepancy between the estimated covariance matrix and the true one in the forecast period, which reduced the benefit of explicitly accounting for covariance uncertainty.

For the Walmart sales, the validation procedure for selecting the bootstrap parameter yielded $\alpha = 0.9$. Our method achieved the highest forecast accuracy for all levels, with substantial performance gains over existing methods.

On the Swiss electricity demand datasets, as determined by validation, the bootstrap parameter was set to $\alpha = 0.7$. Our proposed method achieved the highest forecast accuracy at the upper levels, but the accuracy deteriorated at the bottom level compared to the base forecast. In other words, the improvement at the upper levels came at the expense of accuracy at the bottom level. As with the Australian tourism data, the discrepancy between the true and estimated covariance matrices may have been small, limiting the overall impact of robustification.

Table 6: Forecast accuracy for Australian tourism data

(a) Relative MAE

|        | Base  | BU              | TD              | GLS             | MinT            | Robust          |
|--------|-------|-----------------|-----------------|-----------------|-----------------|-----------------|
| total  | 1.000 | 2.315           | 1.000           | 1.125           | 0.663           | 0.518    |
| state  | 1.000 | $1.277 \pm 0.247$ | $0.968 \pm 0.400$ | $0.635 \pm 0.181$ | $\underline{0.573 \pm 0.135}$ | $0.793 \pm 0.460$ |
| zone   | 1.000 | $1.000 \pm 0.000$ | $0.925 \pm 0.380$ | $0.623 \pm 0.166$ | $\underline{0.568 \pm 0.158}$ | $0.843 \pm 0.387$ |

(b) Relative RMSE

|        | Base  | BU              | TD              | GLS             | MinT            | Robust          |
|--------|-------|-----------------|-----------------|-----------------|-----------------|-----------------|
| total  | 1.000 | 2.132           | 1.000           | 1.105           | 0.698           | 0.519    |
| state  | 1.000 | $1.227 \pm 0.193$ | $0.989 \pm 0.388$ | $0.680 \pm 0.185$ | $\underline{0.617 \pm 0.135}$ | $0.810 \pm 0.428$ |
| zone   | 1.000 | $1.000 \pm 0.000$ | $0.959 \pm 0.367$ | $0.657 \pm 0.154$ | $\underline{0.606 \pm 0.141}$ | $0.844 \pm 0.339$ |

Table 7: Forecast accuracy for Walmart sales data

(a) Relative MAE

|        | Base  | BU              | TD              | GLS             | MinT            | Robust          |
|--------|-------|-----------------|-----------------|-----------------|-----------------|-----------------|
| total  | 1.000 | 1.135           | 1.000           | 1.011           | 0.966           | 0.810    |
| state  | 1.000 | $1.123 \pm 0.117$ | $1.263 \pm 0.986$ | $0.999 \pm 0.029$ | $0.956 \pm 0.049$ | $\underline{0.801 \pm 0.036}$ |
| store  | 1.000 | $1.000 \pm 0.000$ | $1.618 \pm 1.001$ | $0.882 \pm 0.069$ | $0.868 \pm 0.076$ | $\underline{0.741 \pm 0.117}$ |

(b) Relative RMSE

|        | Base  | BU              | TD              | GLS             | MinT            | Robust          |
|--------|-------|-----------------|-----------------|-----------------|-----------------|-----------------|
| total  | 1.000 | 1.129           | 1.000           | 1.011           | 0.967           | 0.817    |
| state  | 1.000 | $1.118 \pm 0.116$ | $1.241 \pm 0.948$ | $0.998 \pm 0.030$ | $0.955 \pm 0.050$ | $\underline{0.807 \pm 0.038}$ |
| store  | 1.000 | $1.000 \pm 0.000$ | $1.580 \pm 0.945$ | $0.885 \pm 0.067$ | $0.872 \pm 0.070$ | $\underline{0.751 \pm 0.111}$ |

Table 8: Forecast accuracy for Swiss electricity demand data

(a) Relative MAE

|          | Base  | BU              | TD              | GLS             | MinT            | Robust          |
|----------|-------|-----------------|-----------------|-----------------|-----------------|-----------------|
| grid     | 1.000 | 1.041           | 1.000           | 1.022           | 1.110           | 0.888    |
| agg.     | 1.000 | $1.045 \pm 0.034$ | $0.983 \pm 0.422$ | $0.983 \pm 0.020$ | $0.995 \pm 0.017$ | $\underline{0.808 \pm 0.277}$ |
| sub-agg. | 1.000 | $1.072 \pm 0.071$ | $1.390 \pm 0.549$ | $0.999 \pm 0.010$ | $1.014 \pm 0.099$ | $\underline{0.867 \pm 0.232}$ |
| meter    | 1.000 | $1.000 \pm 0.000$ | $1.810 \pm 0.921$ | $\underline{0.987 \pm 0.102}$ | $1.023 \pm 0.177$ | $1.036 \pm 0.275$ |

(b) Relative RMSE

|          | Base  | BU              | TD              | GLS             | MinT            | Robust          |
|----------|-------|-----------------|-----------------|-----------------|-----------------|-----------------|
| grid     | 1.000 | 1.022           | 1.000           | 1.012           | 1.050           | 0.946    |
| agg.     | 1.000 | $1.013 \pm 0.045$ | $0.970 \pm 0.213$ | $0.986 \pm 0.008$ | $1.005 \pm 0.023$ | $\underline{0.895 \pm 0.171}$ |
| sub-agg. | 1.000 | $1.032 \pm 0.058$ | $1.170 \pm 0.359$ | $0.995 \pm 0.003$ | $1.025 \pm 0.071$ | $\underline{0.929 \pm 0.158}$ |
| meter    | 1.000 | $1.000 \pm 0.000$ | $1.717 \pm 0.781$ | $\underline{0.981 \pm 0.086}$ | $1.021 \pm 0.141$ | $1.013 \pm 0.227$ |

The numerical experiments on these four datasets demonstrated the effectiveness of our proposed hierarchical time-series forecasting method. Across most target series and hierarchical levels, the proposed approach achieved higher forecast accuracy than existing techniques. However, in a few cases, the proposed method's prediction performance was slightly inferior to that of the GLS and MinT reconciliation methods.

As discussed in the Australian tourism data results, when the discrepancy between the estimated and true covariance matrices is small, incorporating the uncertainty set may lead to overly conservative forecasts, resulting in lower predictive performance. This is further supported by the parameter selection during validation: a larger $\alpha$ was chosen for the Australian births and Walmart sales datasets, where our method was effective. Conversely, a smaller $\alpha$ was selected for the Australian tourism and Swiss electricity demand datasets, where the method was less effective. This suggests that for these latter datasets, the estimated covariance matrix closely approximated the true covariance matrix, reducing the need to account for its uncertainty in the reconciliation process.

## 5 Conclusion

In this paper, we proposed a robust hierarchical time-series forecasting method. This approach introduces an uncertainty set for the inverse of covariance matrix of forecast errors and minimizes the forecast error between the observation values and the coherent forecasts. Through numerical experiments, we demonstrated that the proposed method often provides more accurate forecasts than existing hierarchical time-series forecasting methods, although its performance can vary across different datasets.

The limitations of our method are twofold: its accuracy may not always surpass existing methods, and its scalability is limited due to the computational demands of the optimization problem. The first issue is as stated in the discussion of the experimental results. It arises when the discrepancy between the true and estimated covariance matrices is small, in which case the robust approach offers little advantage. The second limitation concerns the computational time required to obtain a reconciliation matrix, as our method relies on solving a semidefinite optimization problem. The size of the optimization problem depends on the total number of series and the length of observation periods. While a solution was achievable in tens of seconds for the dataset sizes used in our experiments, this approach would not be practical for very large-scale predictions.

### Acknowledgments

This study was conducted as a part of the Data Analysis Competition hosted by the Joint Association Study Group of Management Science. The authors would like to thank the organizers and Research and Innovation Co., Ltd.

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

# A    Full results of numerical experiments

This section reports the original MAE and RMSE for all series from the numerical experiments that could not be included in Section 4.3. Table 9 and Table 10 present the MAE and RMSE for the Australian births data, respectively. Similarly, Table 11 and Table 12 report the forecast accuracy for the Australian tourism data, Table 13 and Table 14 for the Walmart sales data, and Table 15 and Table 16 for the Swiss electricity demand data. Within each table, series separated by horizontal lines correspond to the same hierarchical level.

Table 9: MAE for Australian births data

|       | Base | BU | TD | GLS | MinT | Robust |
|-------|------|------|------|------|------|--------|
| total | 3697.93 | 3822.66 | 3697.93 | 3710.04 | 3695.48 | 3413.01 |
| ACT | 79.34 | 79.34 | 23.95 | 65.71 | 77.18 | 71.65 |
| AUS | 1863.94 | 1863.94 | 1848.74 | 1851.83 | 1819.48 | 1706.96 |
| NSW | 433.36 | 433.36 | 934.11 | 423.74 | 410.68 | 372.41 |
| NT | 41.86 | 41.86 | 43.48 | 30.60 | 39.88 | 36.31 |
| QLD | 222.39 | 222.39 | 231.03 | 217.53 | 217.92 | 202.75 |
| SA | 107.33 | 107.33 | 328.02 | 97.16 | 100.02 | 83.49 |
| TAS | 40.31 | 40.31 | 163.39 | 49.53 | 43.70 | 55.08 |
| VIC | 711.71 | 711.71 | 430.98 | 699.45 | 700.14 | 675.65 |
| WA | 479.92 | 479.92 | 180.03 | 466.29 | 468.86 | 451.56 |

Table 10: RMSE for Australian births data

|       | Base | BU | TD | GLS | MinT | Robust |
|-------|------|------|------|------|------|--------|
| total | 6419.88 | 6507.75 | 6419.88 | 6428.59 | 6418.09 | 6213.25 |
| ACT | 93.99 | 93.99 | 44.91 | 81.93 | 92.04 | 87.08 |
| AUS | 3220.69 | 3220.69 | 3209.85 | 3211.97 | 3188.33 | 3106.97 |
| NSW | 782.22 | 782.22 | 1165.31 | 774.65 | 764.60 | 736.15 |
| NT | 55.47 | 55.47 | 57.62 | 45.46 | 53.67 | 50.50 |
| QLD | 452.41 | 452.41 | 462.21 | 447.36 | 447.76 | 433.83 |
| SA | 166.40 | 166.40 | 354.25 | 157.74 | 160.18 | 146.77 |
| TAS | 54.89 | 54.89 | 169.39 | 62.80 | 57.77 | 67.82 |
| VIC | 1216.24 | 1216.24 | 1004.19 | 1207.09 | 1207.60 | 1190.46 |
| WA | 626.98 | 626.98 | 383.23 | 615.56 | 617.71 | 603.22 |

Table 11: MAE for Australian tourism data

|  | Base | BU | TD | GLS | MinT | Robust |
|---|---|---|---|---|---|---|
| total | 1507.73 | 3490.86 | 1507.73 | 1696.30 | 999.98 | 780.57 |
| NSW | 706.38 | 1056.55 | 336.42 | 594.78 | 445.01 | 678.46 |
| NT | 122.88 | 122.99 | 151.28 | 57.50 | 82.99 | 71.12 |
| QLD | 565.88 | 626.18 | 503.91 | 427.08 | 206.32 | 219.97 |
| SA | 213.29 | 278.99 | 90.94 | 82.64 | 83.73 | 78.47 |
| TAS | 132.18 | 132.50 | 158.77 | 67.01 | 84.52 | 62.70 |
| VIC | 509.36 | 828.63 | 532.14 | 416.27 | 347.01 | 640.93 |
| WA | 331.08 | 462.09 | 498.96 | 222.41 | 207.11 | 503.88 |
| NSW_ACT | 131.54 | 131.54 | 68.63 | 63.37 | 68.05 | 102.13 |
| NSW_Nth | 139.01 | 139.01 | 92.79 | 75.93 | 82.49 | 57.30 |
| NSW_Sth | 131.20 | 131.20 | 115.03 | 77.85 | 94.87 | 165.99 |
| NSW_Metro | 313.69 | 313.69 | 161.47 | 241.31 | 142.86 | 323.77 |
| NSW_Nth_Coast | 237.56 | 237.56 | 119.50 | 160.60 | 130.42 | 190.16 |
| NSW_Sth_Coast | 118.41 | 118.41 | 136.72 | 53.19 | 44.83 | 110.12 |
| NT_Central | 72.85 | 72.85 | 84.78 | 37.56 | 51.82 | 32.97 |
| NT_Nth_Coast | 50.14 | 50.14 | 67.35 | 30.65 | 31.72 | 76.22 |
| QLD_Metro | 360.78 | 360.78 | 209.51 | 318.68 | 197.90 | 173.06 |
| QLD_Central_Coast | 53.20 | 53.20 | 69.45 | 33.78 | 38.48 | 60.19 |
| QLD_Inland | 150.29 | 150.29 | 172.18 | 107.54 | 64.82 | 84.05 |
| QLD_Nth_Coast | 80.10 | 80.10 | 140.75 | 58.20 | 63.47 | 128.42 |
| SA_Metro | 127.55 | 127.55 | 34.11 | 76.69 | 36.13 | 38.15 |
| SA_Inland | 78.95 | 78.95 | 30.53 | 34.33 | 32.78 | 38.62 |
| SA_West_Coast | 27.14 | 27.14 | 24.60 | 31.01 | 16.42 | 22.17 |
| SA_Sth_Coast | 52.22 | 52.22 | 70.85 | 37.05 | 40.02 | 57.76 |
| TAS_Nth_East | 50.76 | 50.76 | 61.78 | 29.15 | 32.34 | 39.28 |
| TAS_Sth | 53.34 | 53.34 | 70.71 | 39.37 | 41.03 | 46.34 |
| TAS_Nth_West | 39.50 | 39.50 | 33.94 | 23.07 | 25.23 | 21.32 |
| VIC_Nth_West | 100.44 | 100.44 | 64.15 | 53.47 | 53.19 | 59.12 |
| VIC_Nth_East | 209.01 | 209.01 | 86.61 | 127.61 | 98.84 | 87.13 |
| VIC_Metro | 335.23 | 335.23 | 339.66 | 250.59 | 176.29 | 335.11 |
| VIC_East_Coast | 113.59 | 113.59 | 126.28 | 73.29 | 79.66 | 154.80 |
| VIC_West_Coast | 85.57 | 85.57 | 93.60 | 39.71 | 43.26 | 59.43 |
| WA_West_Coast | 237.14 | 237.14 | 326.91 | 171.87 | 199.24 | 392.71 |
| WA_Sth | 96.10 | 96.10 | 56.24 | 30.70 | 30.76 | 70.90 |
| WA_Nth | 130.13 | 130.13 | 115.82 | 50.24 | 35.39 | 57.09 |

Table 12: RMSE for Australian tourism data

|  | Base | BU | TD | GLS | MinT | Robust |
|---|---|---|---|---|---|---|
| total | 1726.07 | 3680.00 | 1726.07 | 1906.48 | 1204.56 | 895.21 |
| NSW | 814.62 | 1153.92 | 390.45 | 709.89 | 556.17 | 767.23 |
| NT | 135.97 | 136.13 | 177.71 | 62.33 | 101.17 | 90.27 |
| QLD | 633.93 | 691.06 | 581.55 | 503.91 | 264.39 | 267.44 |
| SA | 226.56 | 291.26 | 105.01 | 98.52 | 99.75 | 94.40 |
| TAS | 148.29 | 148.68 | 185.51 | 86.92 | 96.55 | 78.14 |
| VIC | 642.69 | 938.29 | 697.81 | 553.19 | 475.28 | 725.78 |
| WA | 362.15 | 481.64 | 513.88 | 272.77 | 233.58 | 567.47 |
| NSW_ACT | 146.72 | 146.72 | 86.05 | 79.23 | 83.41 | 117.14 |
| NSW_Nth | 158.32 | 158.32 | 110.01 | 93.05 | 99.64 | 73.21 |
| NSW_Sth | 145.49 | 145.49 | 128.61 | 90.52 | 104.94 | 186.20 |
| NSW_Metro | 370.82 | 370.82 | 192.91 | 300.52 | 178.58 | 365.35 |
| NSW_Nth_Coast | 265.32 | 265.32 | 145.86 | 193.64 | 162.47 | 222.41 |
| NSW_Sth_Coast | 131.95 | 131.95 | 165.04 | 64.79 | 54.05 | 122.21 |
| NT_Central | 87.08 | 87.08 | 102.50 | 48.88 | 69.71 | 41.73 |
| NT_Nth_Coast | 58.57 | 58.57 | 81.15 | 36.06 | 43.98 | 81.02 |
| QLD_Metro | 432.75 | 432.75 | 278.40 | 389.27 | 243.99 | 207.95 |
| QLD_Central_Coast | 62.53 | 62.53 | 78.83 | 42.89 | 48.47 | 64.68 |
| QLD_Inland | 174.46 | 174.46 | 194.94 | 131.45 | 78.24 | 109.60 |
| QLD_Nth_Coast | 101.77 | 101.77 | 183.26 | 73.57 | 70.65 | 145.45 |
| SA_Metro | 134.08 | 134.08 | 41.14 | 84.94 | 45.04 | 43.86 |
| SA_Inland | 87.19 | 87.19 | 36.99 | 43.59 | 42.36 | 46.13 |
| SA_West_Coast | 31.63 | 31.63 | 33.36 | 35.80 | 21.76 | 31.26 |
| SA_Sth_Coast | 66.34 | 66.34 | 82.00 | 42.47 | 46.66 | 69.52 |
| TAS_Nth_East | 58.96 | 58.96 | 70.44 | 35.35 | 39.60 | 47.58 |
| TAS_Sth | 61.49 | 61.49 | 84.42 | 47.51 | 48.14 | 55.55 |
| TAS_Nth_West | 44.88 | 44.88 | 42.50 | 30.26 | 33.01 | 25.17 |
| VIC_Nth_West | 124.89 | 124.89 | 86.11 | 76.51 | 74.90 | 81.43 |
| VIC_Nth_East | 232.47 | 232.47 | 108.39 | 153.51 | 116.29 | 106.73 |
| VIC_Metro | 385.23 | 385.23 | 397.61 | 307.86 | 238.76 | 370.23 |
| VIC_East_Coast | 142.78 | 142.78 | 164.21 | 87.36 | 97.22 | 177.92 |
| VIC_West_Coast | 101.05 | 101.05 | 122.62 | 43.97 | 50.62 | 68.44 |
| WA_West_Coast | 275.32 | 275.32 | 364.37 | 214.80 | 225.39 | 446.61 |
| WA_Sth | 101.13 | 101.13 | 63.93 | 41.77 | 42.26 | 79.92 |
| WA_Nth | 135.48 | 135.48 | 134.23 | 63.10 | 48.77 | 66.16 |

Table 13: MAE for Walmart sales data

|        | Base    | BU      | TD      | GLS     | MinT    | Robust  |
|--------|---------|---------|---------|---------|---------|---------|
| total  | 2343.98 | 2659.41 | 2343.98 | 2370.30 | 2264.97 | 1898.06 |
| CA     | 1237.39 | 1397.73 | 473.01 | 1248.40 | 1195.54 | 993.95  |
| TX     | 586.12  | 587.64  | 1364.16 | 566.09  | 529.02  | 447.99 |
| WI     | 551.73  | 682.22  | 595.33  | 563.94  | 551.06  | 460.93 |
| CA_1   | 217.28  | 217.28  | 509.66  | 182.14  | 196.40  | 169.73 |
| CA_2   | 707.61  | 707.61  | 626.00  | 670.27  | 556.24  | 426.06 |
| CA_3   | 205.31  | 205.31  | 371.79  | 167.98 | 193.55  | 174.84  |
| CA_4   | 268.06  | 268.06  | 153.50 | 230.73  | 250.44  | 224.32  |
| TX_1   | 139.57  | 139.57  | 496.24  | 132.39  | 122.58  | 95.30 |
| TX_2   | 210.50  | 210.50  | 505.23  | 203.52  | 190.62  | 161.45 |
| TX_3   | 239.12  | 239.12  | 362.70  | 232.15  | 216.86  | 193.64 |
| WI_1   | 266.21  | 266.21  | 60.84 | 226.78  | 197.83  | 129.05  |
| WI_2   | 178.76  | 178.76  | 171.12  | 139.94  | 135.31 | 138.43  |
| WI_3   | 238.70  | 238.70  | 456.46  | 199.22  | 220.13  | 195.16 |

Table 14: RMSE for Walmart sales data

|        | Base    | BU      | TD      | GLS     | MinT    | Robust  |
|--------|---------|---------|---------|---------|---------|---------|
| total  | 2479.68 | 2799.50 | 2479.68 | 2507.51 | 2398.58 | 2026.11 |
| CA     | 1315.76 | 1466.86 | 502.11 | 1323.84 | 1267.95 | 1076.92 |
| TX     | 613.93  | 615.60  | 1386.53 | 592.66  | 553.26  | 469.26 |
| WI     | 586.42  | 724.49  | 635.82  | 599.93  | 586.67  | 491.14 |
| CA_1   | 228.58  | 228.58  | 516.72  | 191.94  | 206.43  | 178.76 |
| CA_2   | 761.58  | 761.58  | 630.90  | 727.71  | 623.03  | 518.46 |
| CA_3   | 212.33  | 212.33  | 377.04  | 174.67 | 200.11  | 180.73  |
| CA_4   | 278.59  | 278.59  | 163.45 | 241.39  | 260.40  | 233.43  |
| TX_1   | 147.41  | 147.41  | 503.43  | 139.73  | 129.54  | 101.01 |
| TX_2   | 219.32  | 219.32  | 511.96  | 211.67  | 198.63  | 168.66 |
| TX_3   | 250.99  | 250.99  | 371.73  | 243.43  | 227.41  | 202.47 |
| WI_1   | 283.85  | 283.85  | 90.82 | 241.77  | 211.29  | 136.84  |
| WI_2   | 191.39  | 191.39  | 183.70  | 151.72  | 147.07 | 151.58  |
| WI_3   | 252.62  | 252.62  | 464.93  | 211.08  | 233.04  | 207.25 |

Table 15: MAE for Swiss electricity demand data

|        | Base   | BU     | TD     | GLS    | MinT   | Robust |
|--------|--------|--------|--------|--------|--------|--------|
| grid   | 39.471 | 41.091 | 39.471 | 40.343 | 43.814 | 35.068 |
| S1     | 18.902 | 19.298 | 24.210 | 18.845 | 19.039 | 18.977 |
| S2     | 32.668 | 34.938 | 22.365 | 31.639 | 32.098 | 19.997 |
| S11    | 7.783  | 8.606  | 16.944 | 7.880  | 7.647  | 7.629 |
| S12    | 11.358 | 11.356 | 14.279 | 11.362 | 11.437 | 11.348 |
| S21    | 24.839 | 25.529 | 22.372 | 24.545 | 22.737 | 12.900 |
| S22    | 8.388  | 9.680  | 10.263 | 8.347  | 9.649  | 8.115 |
| S11_1  | 1.087  | 1.087  | 1.864  | 0.783  | 0.478 | 0.489 |
| S11_2  | 2.265  | 2.265  | 4.394  | 2.627  | 2.666  | 2.000 |
| S11_3  | 1.710 | 1.710 | 2.321  | 2.078  | 2.142  | 1.877 |
| S11_4  | 3.008  | 3.008  | 3.335  | 2.968  | 2.909 | 2.994 |
| S11_5  | 6.053  | 6.053  | 15.819 | 5.771  | 5.806  | 5.135 |
| S11_6  | 2.619  | 2.619  | 3.602  | 2.724  | 2.753  | 2.616 |
| S12_1  | 4.558 | 4.558 | 8.822  | 4.563  | 4.760  | 5.088 |
| S12_2  | 1.758  | 1.758  | 1.667 | 1.756  | 2.023  | 2.144 |
| S12_3  | 4.519  | 4.519  | 7.814  | 4.515 | 5.325  | 4.572 |
| S12_4  | 4.263  | 4.263  | 6.355  | 4.268  | 4.233  | 3.121 |
| S12_5  | 4.499  | 4.499  | 4.249  | 4.504  | 4.169 | 5.501 |
| S12_6  | 2.505  | 2.505  | 2.161 | 2.502  | 2.884  | 2.610 |
| S21_1  | 1.359  | 1.359  | 2.935  | 1.470  | 1.174 | 1.490 |
| S21_2  | 1.124  | 1.124  | 1.135  | 1.180  | 1.106 | 1.297 |
| S21_3  | 0.872  | 0.872  | 2.793  | 0.848  | 0.841 | 1.385 |
| S21_4  | 2.751  | 2.751  | 10.583 | 2.812  | 2.745 | 3.352 |
| S21_5  | 6.562  | 6.562  | 4.230 | 6.398  | 7.161  | 5.460 |
| S21_6  | 17.541 | 17.541 | 18.843 | 17.377 | 13.768 | 9.224 |
| S22_1  | 2.173  | 2.173  | 5.764  | 1.866 | 2.493  | 2.296 |
| S22_2  | 1.928  | 1.928  | 7.793  | 1.687 | 2.118  | 2.091 |
| S22_3  | 3.880  | 3.880  | 8.993  | 3.711  | 3.702 | 4.317 |
| S22_4  | 1.100  | 1.100  | 1.983  | 1.198  | 1.093 | 1.101 |
| S22_5  | 1.225  | 1.225  | 1.180  | 1.088  | 1.266  | 1.066 |
| S22_6  | 1.168  | 1.168  | 1.998  | 0.994 | 1.581  | 2.003 |

Table 16: RMSE for Swiss electricity demand data

|       | Base   | BU     | TD     | GLS    | MinT   | Robust |
|-------|--------|--------|--------|--------|--------|--------|
| grid  | 64.301 | 65.685 | 64.301 | 65.102 | 67.498 | 60.821 |
| S1    | 30.940 | 30.341 | 34.684 | 30.683 | 31.597 | 31.432 |
| S2    | 39.665 | 41.439 | 32.503 | 38.863 | 39.198 | 30.729 |
| S11   | 12.023 | 11.953 | 20.456 | 11.951 | 12.321 | 12.442 |
| S12   | 19.037 | 18.966 | 17.871 | 18.976 | 19.450 | 19.160 |
| S21   | 28.347 | 28.919 | 27.276 | 28.105 | 26.616 | 19.665 |
| S22   | 11.817 | 13.200 | 12.733 | 11.781 | 13.164 | 11.596 |
| S11_1 | 1.166  | 1.166  | 2.088  | 0.877  | 0.649 | 0.656  |
| S11_2 | 2.800  | 2.800  | 5.079  | 3.117  | 3.152  | 2.526 |
| S11_3 | 1.968 | 1.968 | 2.775  | 2.304  | 2.365  | 2.111  |
| S11_4 | 3.466  | 3.466  | 4.334  | 3.409 | 3.596  | 3.862  |
| S11_5 | 6.399  | 6.399  | 16.354 | 6.105  | 6.142  | 5.449 |
| S11_6 | 4.682  | 4.682  | 5.583  | 4.823  | 4.850  | 4.666 |
| S12_1 | 6.568  | 6.568  | 12.222 | 6.571  | 6.566 | 6.621  |
| S12_2 | 2.350  | 2.350  | 2.105 | 2.348  | 2.586  | 2.688  |
| S12_3 | 5.037  | 5.037  | 8.329  | 5.034 | 5.782  | 5.077  |
| S12_4 | 4.726  | 4.726  | 7.384  | 4.730  | 4.707  | 4.183 |
| S12_5 | 5.247  | 5.247  | 5.082  | 5.252  | 5.062 | 6.016  |
| S12_6 | 2.744  | 2.744  | 2.989  | 2.741 | 3.099  | 2.844  |
| S21_1 | 1.739  | 1.739  | 3.783  | 1.800  | 1.655 | 1.811  |
| S21_2 | 1.566  | 1.566  | 1.614  | 1.592  | 1.560 | 1.648  |
| S21_3 | 1.022  | 1.022  | 3.715  | 0.996  | 0.993 | 1.625  |
| S21_4 | 4.406  | 4.406  | 11.238 | 4.360  | 4.415  | 4.228 |
| S21_5 | 7.181  | 7.181  | 5.095 | 7.034  | 7.725  | 6.216  |
| S21_6 | 18.548 | 18.548 | 19.619 | 18.400 | 15.199 | 11.585 |
| S22_1 | 2.559  | 2.559  | 6.203  | 2.256 | 2.858  | 2.672  |
| S22_2 | 2.339  | 2.339  | 8.260  | 2.046 | 2.590  | 2.557  |
| S22_3 | 6.345  | 6.345  | 10.247 | 6.159  | 5.731  | 5.548 |
| S22_4 | 1.326  | 1.326  | 2.465  | 1.443  | 1.310 | 1.320  |
| S22_5 | 1.717  | 1.717  | 1.497  | 1.522  | 1.758  | 1.484 |
| S22_6 | 1.424  | 1.424  | 2.417  | 1.226 | 1.838  | 2.248  |

