# OpenReview forum: "Hierarchical Time Series Forecasting with Robust Reconciliation"
_TMLR — Accepted by TMLR_

### Review · Reviewer_DVTz · 2025-11-03

**Summary Of Contributions:**

The authors suggest using a robust optimization procedure for the reconciliation of hierarchical time-series forecasts. The authors start with the MinT method proposed by Wickramasuriya et al. (2019). They modify an optimization problem to a robust optimization problem to account for errors in covariance matrix estimation. The efficiency of the proposed method is demonstrated in experiments with time-series data.

**Strengths**
1. The motivation for the underlying problem is clear. Time-series forecasting is an important problem for practical applications.
2. The writing is clear.
3. The proposed approach seems novel.

**Weaknesses**
1. The motivation for the exact research question of the paper is vague (see below).
2. The paragraph about the limitations of the method at the end of Section 4.3 does not directly lead to a practical rule to choose between MinT and the new method.

**Audience:**

Yes

**Audience Explanation:**

I think the paper might be of interest to communities that perform time-series forecasting.

**Broader Impact Concerns:**

I am not concerned about the broader ethical implications of the work.

**Claims And Evidence:**

No

**Claims Explanation:**

1. I am confused about the optimization problem. Panagiotelis et al. (2021) indeed showed that the MinT method is equivalent to the optimization of Equation (3). However, the canonical weight matrix in this problem is $W = (W^{(\tau)})^{-1}$, not $W = W^{(\tau)}$.
2. Similarly, Theorem 3.3 in Panagiotelis et al. (2021) shows that the MinT solution is optimal for any matrix. It suggests that the exact choice of matrix $W$ might not be so crucial for the optimization. Thus, the rationale for applying robust methods to the matrix $W$ is unclear.
3. The current theoretical results in Section 3 do not demonstrate the advantage of the proposed method in terms of expected error.

**Requested Changes:**

1. Please explain the motivation behind the approach.
2. Explain why the expected error is supposed to decrease.
3. (minor) Please expand on the limitations and provide a practical guide for using your method.

---

> ### Author Response · Authors · 2025-11-28
> **Reply 1 to the comment by Reviewer DVTz**
>
> Thank you for your careful review and insightful comments.
> We are grateful for your constructive feedback.
> We will incorporate all reviewers' feedback and additional experimental results into the revised manuscript.
>
> We provide our point-by-point responses to your comments below.
>
> > I am confused about the optimization problem. Panagiotelis et al. (2021) indeed showed that the MinT method is equivalent to the optimization of Equation (3). However, the canonical weight matrix in this problem is $W=(W^{(\tau)})^{-1}$, not $W=W^{(\tau)}$.
>
> We sincerely thank you for this crucial observation.
> As you correctly pointed out, the canonical weight matrix should be the inverse of the covariance matrix ($W=(W^{(\tau)})^{-1}$), whereas we had incorrectly applied the covariance matrix itself ($W=W^{(\tau)}$) in the original manuscript.
> We have corrected the formulation in the revised manuscript to use the inverse matrix.
> Accordingly, we re-conducted all numerical experiments using the correct weight matrix.
> We confirmed that the results exhibit the same trends as those originally reported, demonstrating that the proposed method continues to achieve high predictive accuracy.
> The relative MAE results from the re-experiments are presented below:
>
> - Results for Australian birth data
>
> |               | Base           | BU            | TD               | GLS              | MinT             | Robust            |
> |---------------|----------------|---------------|------------------|------------------|------------------|-------------------|
> | total         | 1.000          | 1.027         | 1.000            | 1.003            | 1.003            | **0.929**             |
> | state         | 1.000 ± 0.000  | 1.000 ± 0.000 | 1.518 ± 1.224    | 0.963 ± 0.104    | 0.985 ± 0.031    | **0.949 ± 0.146**     |
>
> - Results for Australian tourism data
>
> |           | Base           | BU                        | TD                        | GLS                       | MinT                      | Robust                    |
> |-----------|----------------|----------------------------|----------------------------|----------------------------|----------------------------|---------------------------|
> | total     | 1.000          | 2.315                      | 1.000                      | 1.125                      | 0.663                      | **0.518**                     |
> | state     | 1.000 ± 0.000  | 1.277 ± 0.229              | 0.968 ± 0.371              | 0.635 ± 0.168              | **0.573 ± 0.125**              | 0.793 ± 0.426             |
> | zone      | 1.000 ± 0.000  | 1.000 ± 0.000              | 0.925 ± 0.373              | 0.623 ± 0.163              | **0.568 ± 0.155**              | 0.843 ± 0.380             |
>
> - Results for Walmart sales data
>
> |               | Base           | BU              | TD               | GLS              | MinT             | Robust            |
> |---------------|----------------|------------------|------------------|------------------|------------------|-------------------|
> | total         | 1.000          | 1.141            | 1.000            | 1.012            | 0.964            | **0.884**             |
> | state           | 1.000 ± 0.000  | 1.132 ± 0.110    | 1.260 ± 0.800    | 1.000 ± 0.027    | 0.953 ± 0.045    | **0.864 ± 0.027**     |
> | store         | 1.000 ± 0.000  | 1.000 ± 0.000    | 1.601 ± 0.929    | 0.878 ± 0.071    | 0.859 ± 0.085    | **0.791 ± 0.145**     |
>
> - Results for Swiss electricity demand data
>
> |           | Base        | BU                       | TD                       | GLS                       | MinT                      | Robust                    |
> |-----------|-------------|---------------------------|---------------------------|---------------------------|---------------------------|---------------------------|
> | grid      | 1.000       | 1.050                     | 1.000                     | 1.020                     | 1.119                     | **0.893**                     |
> | agg.      | 1.000 ± 0.000 | 1.052 ± 0.038            | 0.990 ± 0.294             | 0.988 ± 0.009             | 1.004 ± 0.004             | **0.817 ± 0.187**             |
> | sub-agg.  | 1.000 ± 0.000 | 1.072 ± 0.066            | 1.393 ± 0.485             | 0.996 ± 0.013             | 1.016 ± 0.086             | **0.871 ± 0.200**             |
> | meter     | 1.000 ± 0.000 | 1.000 ± 0.000            | 1.809 ± 0.895             | **0.984 ± 0.102**             | 1.023 ± 0.171             | 1.039 ± 0.265             |

---

> ### Author Response · Authors · 2025-11-28
> **Reply 2 to the comment by Reviewer DVTz**
>
> > Similarly, Theorem 3.3 in Panagiotelis et al. (2021) shows that the MinT solution is optimal for any matrix. It suggests that the exact choice of matrix $W$ might not be so crucial for the optimization.  Thus, the rationale for applying robust methods to the matrix $W$ is unclear.
>
> Thank you for your important comment.
> We would like to clarify a fundamental difference in the minimization objectives between MinT and our proposed method.
> Previous studies, including Panagiotelis et al. (2021), aim to minimize the expected value (variance) of forecast errors under specific probabilistic assumptions about the error terms.
> In contrast, our study minimizes the residuals (forecast errors) themselves derived from the observed data, without relying on assumptions about the error terms.
> In this formulation of "residual minimization," the weight matrix $W$ directly determines the structure of the loss function.
> We will clarify this difference in the revised version.
>
> > The current theoretical results in Section 3 do not demonstrate the advantage of the proposed method in terms of expected error.
>
> We acknowledge that it is difficult to theoretically prove that our method always improves the expected error under ideal assumptions.
> However, we would like to emphasize that the primary contribution of our formulation lies in its practical value.
> In real-world scenarios where the true covariance matrix is unknown, robustness against estimation errors is often more critical than theoretical optimality under perfect information.
>
> The fact that our method improved predictive performance in the numerical experiments demonstrates that this "uncertainty-aware" formulation is not merely a theoretical concept, but a practically effective approach.
>
>  > Please explain the motivation behind the approach.
>
> Our primary motivation is to address a practical vulnerability in MinT, where performance degrades significantly due to estimation errors of the unknown covariance matrix.
> Our approach aims to achieve stable and high predictive accuracy even when accurate covariance estimation is difficult, by formulating the reconciliation problem to be robust against the uncertainty of the covariance matrix.
>
> > Explain why the expected error is supposed to decrease.
>
> In practical scenarios where the true error distribution is unknown, significant discrepancies often arise between the covariance matrix estimated from training data and the true covariance of unseen data.
> By minimizing the worst-case error over an uncertainty set, the proposed method is designed to prevent severe prediction failures caused by such discrepancies.
> We expect that this robustness consequently leads to an improvement in the overall expected error (e.g., residual sum of squares) compared to MinT or the base forecasts.
>
> > (minor) Please expand on the limitations and provide a practical guide for using your method.
>
> As indicated by our numerical experiments, the performance improvement of our method is marginal when the covariance matrix is estimated with high accuracy (i.e., the discrepancy between the estimated and true matrices is small).
> However, the method becomes particularly effective in scenarios where accurate estimation is difficult, such as time series with irregular fluctuations or time-varying error distributions.
> We acknowledge that developing a quantitative criterion to determine the method's effectiveness a priori based on data characteristics is challenging, and this remains a topic for future work.
>
> The computation time for our optimization problem depends on the number of time series.
> Our experiments confirmed that for datasets with dozens of series (medium-scale), the problem can be solved within a practical time using general-purpose semidefinite programming (SDP) solvers.
> Therefore, we currently recommend applying this method to medium-scale hierarchical data, rather than large-scale datasets with hundreds or thousands of series.

---

### Review · Reviewer_4M2K · 2025-11-05

**Summary Of Contributions:**

The main technical problem in hierarchical time series forecasting is the estimation of the reconciliation matrix that encodes the hierarchical aggregation constrains. In state-of-the-art techniques this reconciliation matrix is obtained by estimating the covariance matrix of the errors between observation values and base forecasts. In this paper they propose a novel robust method that estimates the reconciliation matrix as the min-max solution over an uncertainty set of possible covariance matrices. The resulting min-max optimization problem is solved by formulating it as a semidefinite program. The method is tested in several dataset and results in improved MAE and RMSE metrics for in most of them when compared to state-of-the-art hierarchical time series forecasting methods.

**Additional Comments:**

N/A

**Audience:**

Yes

**Audience Explanation:**

The paper may be interesting for both the time series forecasting and robust optimization communities.

**Claims And Evidence:**

Yes

**Claims Explanation:**

The problem statement, modelling steps and the experimental setup are explained with sufficient detail and clarity.

**Requested Changes:**

Your method is basically is a tool to convert a base forecast into a hierarchical forecast, and in theory, it is agnostic to the base forecasting technique used. In the experiments you mention the use of the Prophet library for base forecasting. Have you compared your method to other reconciliation methods using different base forecasts? In principle, it could happen that different base forecasting techniques interacted in unexpected ways with the different reconciliation methods you analyze. From what you mention in subsection 4.2, it wouldn't be hard to add some extra experiments to verify that your method is consistently superior over different base forecasting techniques. If there are theoretical reasons to expect that, they also deserve some discussion.

---

> ### Author Response · Authors · 2025-11-28
> **Reply 1 to the comment by Reviewer 4M2K**
>
> Thank you for your careful review and insightful comments.
> We are grateful for your constructive feedback. We will incorporate all reviewers' feedback and additional experimental results into the revised manuscript.
>
> Before addressing your specific comments, we would like to mention a correction regarding the experimental settings.
> Another reviewer (Reviewer DVTz) pointed out an error in our formulation of the forecast error minimization problem:
>
> > Panagiotelis et al. (2021) indeed showed that the MinT method is equivalent to the optimization of Equation (3). However, the canonical weight matrix in this problem is $W=(W^{(\tau)})^{-1}$, not $W=W^{(\tau)}$.
>
> As pointed out, the canonical weight matrix should be the inverse of the covariance matrix, $W=(W^{(\tau)})^{-1}$.
> However, in the original manuscript, we directly applied the covariance matrix $W=W^{(\tau)}$.
>
> Following this feedback, we re-ran all numerical experiments using the correct formulation with the inverse covariance matrix.
> We confirmed that the **results show the same trends** as before, and the validity of our proposed method remains unchanged.
> All numerical results and figures in the revised manuscript and this response letter have been updated based on these re-experiments.
> For detailed results of these re-experiments, please refer to our response to Reviewer DVTz.
>
> With this context, we provide our point-by-point responses to your comments below.
>
> > Your method is basically is a tool to convert a base forecast into a hierarchical forecast, and in theory, it is agnostic to the base forecasting technique used. In the experiments you mention the use of the Prophet library for base forecasting. Have you compared your method to other reconciliation methods using different base forecasts? In principle, it could happen that different base forecasting techniques interacted in unexpected ways with the different reconciliation methods you analyze. From what you mention in subsection 4.2, it wouldn't be hard to add some extra experiments to verify that your method is consistently superior over different base forecasting techniques. If there are theoretical reasons to expect that, they also deserve some discussion.
> Thank you for the important comments. As you pointed out, our proposed method is theoretically agnostic to the base forecasting methods, and it is difficult to theoretically prove that our method will consistently improve prediction performance.
>
> Regarding our choice of Prophet in the manuscript: we selected it because preliminary experiments comparing several methods (including ARIMA, XGBoost, and LightGBM) showed that Prophet generally yielded good average performance for the base forecasts.
>
> Following your suggestion to experimentally verify whether the proposed method improves performance regardless of the base forecasting technique used, we conducted additional experiments on the Australian births dataset using ARIMA, XGBoost, and LightGBM.
> The results are summarized in the tables below:

---

> ### Author Response · Authors · 2025-11-28
> **Reply 2 to the comment by Reviewer 4M2K**
>
> - Prophet
>
> |               | Base           | BU            | TD               | GLS              | MinT             | Robust            |
> |---------------|----------------|---------------|------------------|------------------|------------------|-------------------|
> | total         | 1.000          | 1.027         | 1.000            | 1.003            | 1.003            | **0.929**             |
> | state         | 1.000 ± 0.000  | 1.000 ± 0.000 | 1.518 ± 1.224    | 0.963 ± 0.104    | 0.985 ± 0.031    | **0.949 ± 0.146**     |
>
> - ARIMA
>
> |               | Base           | BU            | TD               | GLS              | MinT             | Robust            |
> |---------------|----------------|---------------|------------------|------------------|------------------|-------------------|
> | total         | 1.000          | 0.982         | 1.000            | 0.988            | **0.970**            | 0.978             |
> | state         | 1.000 ± 0.000  | 1.000 ± 0.000 | 1.629 ± 0.979    | 2.275 ± 1.686    | 0.990 ± 0.016    | **0.979 ± 0.141**     |
>
> - XGBoost
>
> |               | Base           | BU            | TD               | GLS              | MinT             | Robust            |
> |---------------|----------------|---------------|------------------|------------------|------------------|-------------------|
> | total         | 1.000          | 1.001         | 1.000            | **1.000**            | 1.001            | 1.003             |
> | state         | **1.000 ± 0.000**  | **1.000 ± 0.000** | 1.915 ± 1.378    | 1.056 ± 0.086    | 1.000 ± 0.006    | 1.001 ± 0.007     |
>
> - LightGBM
>
> |               | Base           | BU            | TD               | GLS              | MinT             | Robust            |
> |---------------|----------------|---------------|------------------|------------------|------------------|-------------------|
> | total         | 1.000          | 0.981         | 1.000            | 0.998            | 0.976            | **0.952**             |
> | state         | 1.000 ± 0.000  | 1.000 ± 0.000 | 1.865 ± 1.584    | 1.177 ± 0.236    | **0.998 ± 0.007**    | 0.998 ± 0.025     |
>
> **Discussion of Results:**
>
> **General Effectiveness:** For Prophet, ARIMA, and LightGBM, the proposed method (Robust) consistently improved upon the base forecasts.
> Furthermore, its performance was comparable to or better than the MinT method.
>
> **The Case of XGBoost:** In the case of XGBoost, we observed that our method did not improve the accuracy.
> However, notably, MinT—which theoretically guarantees improvement under ideal assumptions—also failed to improve upon the Base forecast.
> This suggests that the issue lies not with the reconciliation methods, but likely with the properties of the XGBoost base forecasts themselves (e.g., specific biases or a lack of exploitable correlation structure).
>
> While it is challenging to theoretically guarantee improvement for every base forecasting method, these results demonstrate the merit of our approach: **it performs robust hierarchical reconciliation even when the true covariance matrix is uncertain**.
> By minimizing the worst-case error over an uncertainty set, the proposed method provides stable predictive performance and prevents significant accuracy degradation, demonstrating that this robustness is independent of the specific base forecasting technique employed.

---

### Review · Reviewer_z2xd · 2025-11-17

**Summary Of Contributions:**

In this paper is proposed a way to improve upon the reconciliation method for hierarchical time series problems. Its main starting point is the minT method, which it improves upon. Whereas in minT, the co-variance used for least squares estimation is a single step process consisting of all the samples, in the current method, they take bootstrap samples from the dataset and come up with min and max (or rather, low and high percentile) values of these. They then solve a min-max problem (min P max W Error ), so that it can be performant even under the worst case scenarios of W - this is different from minT which does it over a single pass.

The minmax problem is solved using positive semi-definite programming. I am not very familiar with the methods used so I cannot comment on the SDP approach with confidence.

They carry out evaluations over three different hierarchical time series datasets (Australian tourism, Walmart sales and Swiss electricity demand), and show comparisons with other methods (topdown, bottom up, ordinary least squares, minT). The numbers show that they with the robust method, they are able to obtain lower MAE/RMSE compared to the other methods.

I would like to emphasize that I am not an expert in this area and my confidence estimate of the review is therefore is low.

**Audience:**

Yes

**Audience Explanation:**

This is an important addition to the time series reconciliation problem, overcoming some of the limitations of the well known minT method.

**Claims And Evidence:**

Yes

**Claims Explanation:**

The motivation is very clear in that the method overcomes limitations of minT which can be sensitive to the choice of W. Results are better than minT.

**Requested Changes:**

Could the authors provide some insight into why the method underperforms in the tourism dataset?

---

> ### Author Response · Authors · 2025-11-28
> **Reply to the comment by Reviewer z2xd**
>
> Thank you for your careful review and insightful comments.
> We are grateful for your constructive feedback.
> We will incorporate all reviewers' feedback and additional experimental results into the revised manuscript.
>
> Before addressing your specific comments, we would like to mention a correction regarding the experimental settings.
> Another reviewer (Reviewer DVTz) pointed out an error in our formulation of the forecast error minimization problem:
>
> > Panagiotelis et al. (2021) indeed showed that the MinT method is equivalent to the optimization of Equation (3). However, the canonical weight matrix in this problem is $W=(W^{(\tau)})^{-1}$, not $W=W^{(\tau)}$.
>
> As pointed out, the canonical weight matrix should be the inverse of the covariance matrix, $W=(W^{(\tau)})^{-1}$.
> However, in the original manuscript, we directly applied the covariance matrix $W=W^{(\tau)}$.
>
> Following this feedback, we re-ran all numerical experiments using the correct formulation with the inverse covariance matrix.
> We confirmed that the **results show the same trends** as before, and the validity of our proposed method remains unchanged.
> All numerical results and figures in the revised manuscript and this response letter have been updated based on these re-experiments.
> For detailed results of these re-experiments, please refer to our response to Reviewer DVTz.
>
> With this context, we provide our point-by-point responses to your comments below.
>
> > Could the authors provide some insight into why the method underperforms in the tourism dataset?
>
>  As mentioned earlier, we re-evaluated the Tourism dataset using the corrected formulation (using the inverse covariance matrix).
> The updated results are shown in the table below:
>
> |           | Base           | BU                        | TD                        | GLS                       | MinT                      | Robust                    |
> |-----------|----------------|----------------------------|----------------------------|----------------------------|----------------------------|---------------------------|
> | total     | 1.000          | 2.315                      | 1.000                      | 1.125                      | 0.663                      | **0.518**                     |
> | state     | 1.000 ± 0.000  | 1.277 ± 0.229              | 0.968 ± 0.371              | 0.635 ± 0.168              | **0.573 ± 0.125**              | 0.793 ± 0.426             |
> | zone      | 1.000 ± 0.000  | 1.000 ± 0.000              | 0.925 ± 0.373              | 0.623 ± 0.163              | **0.568 ± 0.155**              | 0.843 ± 0.380             |
>
> As shown in the table, while the proposed method (Robust) achieves the best performance at the aggregate "Total" level, it indeed underperforms compared to MinT and GLS at the "State" and "Zone" levels.
>
> We attribute this result to the likely small divergence between the covariance matrix estimated from the observation period and the true covariance matrix during the prediction period in this specific dataset.
>
> MinT reconciles base forecasts under the assumption that the estimated covariance matrix is accurate.
> In contrast, our proposed method focuses on the "worst-case" scenario, accounting for potential uncertainties or errors in the covariance estimation.
>
> Therefore, in scenarios where the discrepancy between the estimated and true covariance matrices is minimal (i.e., the estimation is accurate), the proposed method may become overly conservative.
> This leads to lower prediction accuracy compared to methods like MinT, which directly uses the estimated matrix without assuming worst-case deviations.
> This suggests that our method is most beneficial in situations where covariance estimation is unstable or subject to distributional shifts.

---

### Decision · Action_Editor_74AB · 2025-12-30

**Recommendation:** Accept as is

**Audience:**

Yes

**Audience Explanation:**

The key contribution of the paper is a robust reconciliation framework that explicitly accounts for uncertainty in the estimation of the forecast error covariance matrix.

**Claims And Evidence:**

Yes

**Claims Explanation:**

This paper addresses the problem of forecast reconciliation in hierarchical time series, where independently generated base forecasts must be adjusted to satisfy aggregation constraints across different hierarchical levels. The work builds on the widely used MinT (Minimum Trace) reconciliation method, which produces optimal reconciled forecasts under the assumption that the covariance matrix of forecast errors is accurately estimated.